# CBP and p300 Jointly Maintain Neural Progenitor Viability but Play Unique Roles in the Differentiation of Neural Lineages

**DOI:** 10.3390/cells11244118

**Published:** 2022-12-18

**Authors:** Rocío González-Martínez, Angel Márquez-Galera, Beatriz Del Blanco, Jose P. López-Atalaya, Angel Barco, Eloísa Herrera

**Affiliations:** Instituto de Neurociencias, Consejo Superior de Investigaciones Científicas-Universidad Miguel Hernández, CSIC-UMH. Av. Santiago Ramón y Cajal s/n, 03550 Alicante, Spain

**Keywords:** neuronal differentiation, neural progenitor proliferation, CBP, p300, lysine acetylation, transcriptomics, epigenomics, epigenetics, intellectual disability, Rubinstein–Taybi syndrome

## Abstract

The paralogous lysine acetyltransferases 3 (KAT3), CBP and P300, play critical roles during neurodevelopment, but their specific roles in neural precursors maintenance and differentiation remain obscure. In fact, it is still unclear whether these proteins are individually or jointly essential in processes such as proliferation of neural precursors, differentiation to specific neural cell types, or both. Here, we use subventricular zone-derived neurospheres as a potential ex vivo developmental model to analyze the proliferation and differentiation of neural stem cells (NSCs) lacking CBP, p300, or both proteins. The results showed that CBP and p300 are not individually essential for maintenance and proliferation of NSCs, although their combined ablation seriously compromised cell division. In turn, the absence of either of the two proteins compromised the differentiation of NSC into the neuronal and astrocytic lineages. Single-nucleus RNA sequencing analysis of neural cell cultures derived from CBP or p300 mutant neurospheres revealed divergent trajectories of neural differentiation upon CBP or p300 ablation, confirming unique functions and nonredundant roles in neural development. These findings contribute to a better understanding of the shared and individual roles of KAT3 proteins in neural differentiation and the etiology of neurodevelopmental disorders caused by their deficiency.

## 1. Introduction

The development of multicellular organisms relies on the differentiation of pluripotent cells into specialized cell types. This is achieved by the coordinated action of transcription factors and epigenetic regulators that introduce epigenetic modifications on the chromatin to promote differentiation and establish cell fates. The prime example of complexity and refinement is the brain, where dozens or possibly hundreds of cellular types interact to generate sophisticated patterns of brain activity. Among the multiple epigenetic mechanisms involved in development, the acetylation and deacetylation of lysine residues of histone tails by lysine acetyltransferases (KATs) and deacetylases (KDACs) play a key role in modulating chromatin structure and enabling or blocking gene expression [1].

The paralogous transcriptional coactivators with intrinsic KAT activity, CBP and p300, are ubiquitously expressed in the nervous system and are essential during neurodevelopment [2,3]. The two proteins are structurally very similar and constitute the KAT3 family of lysine acetyltransferases. In fact, hemi-deficiency of either one of them causes a severe neurodevelopmental disorder known as Rubinstein–Taybi syndrome (RSTS type 1 [OMIM #180849] and 2 OMIM #613684], respectively for CBP and p300). Although the function of these proteins has been extensively studied, the specific developmental events downstream of CBP and p300 deficiencies remain unclear. In vertebrates, these proteins are key for neurulation because mutant mice lacking CBP or p300 are embryonic lethal and show defects in neural tube closure [4,5]. However, neurulation and the subsequent formation of the nervous system are very complex processes involving proliferation, neural differentiation, and neuronal maturation; furthermore, at the cellular level, the consequence of removing KAT3 proteins in the different steps of development is not clear. RNA-seq data have shown that p300 is less expressed in neurons than CBP in both adult and embryonic brain [3]; however, whether they have redundant roles in different contexts remains also unknown.

To precisely elucidate the specific requirement of KAT3 proteins in the proliferation and differentiation of neural stem cells (NSCs), we took advantage of conditional mutant mice for CBP and p300 to generate NSC cultures lacking either one or both proteins. NSCs from the subventricular zone (SVZ), a neurogenic niche at both embryonic and postnatal stages, proliferate to form multipotent clonal aggregates called neurospheres when cultured under appropriate conditions [6]. These neurospheres display self-renewal and proliferation and are able to differentiate into the three main neural linages, neurons, astrocytes, and oligodendrocytes. Using this ex vivo neurosphere model, we analyzed the biological properties of NSCs lacking CBP, p300, or both, and we found that CBP and p300 have a redundant role in NSCs self-renewal. Then, cellular characterization and single-nucleus RNA sequencing analysis of differentiating neurospheres lacking either CBP or p300 underscored the crucial role of both KAT3 proteins in the transcriptional program underlying the establishment of the astrocyte lineage, as well as their involvement in proper differentiation of neurons and oligodendrocytes.

## 2. Materials and Methods

### 2.1. Animals

The generation of *Crebbp*^f/f^ (Zhang et al., 2004) and *Ep300*^f/f^ [7] mice has been previously described. These two strains were crossed to obtain double-knockout mice. The genetic background of all mice was C57BL/6. For developmental studies, embryonic day 0.5 (E0.5) was defined by the day of vaginal plug detection, and day of birth was defined as postnatal day 0 (P0). The genotype of the mice was determined by PCR on DNA extracted from tail tissue using the following primers: CBP floxed allele (wildtype band 230 bp; mutant band 300 bp): fw 5′–CCTCTGAAGGAGAAACAAGCA–3′ and rv 5′–ACCATCATTCATCAGTGGACT–3′; p300 floxed allele (wildtype band 247 bp; mutant band 400 bp): fw 5′–GTGAGTTGATGTCCCTGTCG–3′ and rv 5′–CAGACACCCTCTTGCACTCA–3′. For all PCR reactions, thermocycling conditions were as follows: 95 °C for 4 min followed by 35 cycles of 1 min at 95 °C, 1 min at 54 °C, and 1 min at 72 °C, and a final extension at 72 °C for 10 min. All genotyping reactions contained approximately 200 ng of DNA template, 0.5 μM of each primer, 1.25 mM MgCl_2_, 5 U/µL KAPA Taq DNA Polymerase, and 1× KAPA Taq buffer in a 20 μL final volume. Mice were maintained at the Instituto de Neurociencias in a specific pathogen-free facility with a 12 h light/dark cycle, controlled temperature (22 ± 1 °C), and relative humidity (55 ± 5%) with food and water ad libitum. All animal experiments were approved by and performed in accordance with the protocols of the IN Animal Care and Use Committee and met Spanish (RD 53/2013 and European (2013/63/UE) regulations.

### 2.2. Establishment of a Primary Neural Stem Cell (NSC) Culture

To establish primary neurosphere cultures, the protocol was adapted from previous studies [8,9,10,11] *Crebbp*^f/f^, *Ep300*^f/f^, or *Crebbp*^f/f^x*Ep300*^f/f^ mice pups (P0–P3) were sacrificed by decapitation, and brains were extracted on iced cold Hank’s balanced salt solution (HBSS, Gibco), removing meninges and blood vessels microscopically. Slices of 2–3 mm containing the lateral ventricles were dissected under microscope, removing olfactory bulbs and the rest of the brain caudal to the slices. The subventricular zone (SVZ) was immediately dissected, and then incubated with 12.5 mg/mL trypsin (Sigma) at 37 °C for 5 min, followed by resuspension in HBSS with 10% fetal bovine serum (FBS, Gibco) to block enzymatic activity. SVZs were mechanically dissociated into a cell suspension with a fire-polished Pasteur pipette, centrifuged, and resuspended in NSC complete serum-free medium containing DMEM/F12 with L-glutamine 1× (Gibco) supplemented with 0.6% D-glucose (Panreac), 0.1% NaHCO_3_ (Sigma), 5 mM HEPES (Gibco) 2 mM L-Glutamine (Gibco), 1% penicillin/streptomycin (Gibco), 0.7 U/mL heparin sodium salt (Sigma), 4 mg/mL bovine serum albumin (BSA) (Sigma), 0.8 mg/mL apo-transferrin (Sigma), 500 nM bovine insulin (Sigma), 0.1 mg/mL putrescine (Sigma), 0.2 nM progesterone (Sigma), and 0.3 μM sodium selenite (Sigma), as well as growth factors (10 ng/mL human recombinant basic fibroblast growth factor (bFGF) (Sigma) and 20 ng/mL epidermal growth factor (EGF) (Gibco) as mitogens. Cells were counted in a Neubauer chamber and seeded at a density of 5 × 10^4^ cells/mL in 24-well (0.5 mL/well) uncoated ultralow-attachment plates (Corning^®^) and placed at 37 °C with 5% CO_2_ for 14 days. The day of plating was considered day in vitro 1 (DIV1). To assess self-renewal and proliferation of NSCs, we performed the clonal neurosphere assay (NSA) [9]. Free-floating neurospheres were pooled for enzymatic dissociation with Accutase^®^ (Sigma), a reproducible method to obtain a single-cell suspension with high viability. Approximately neurospheres from 20–24 wells were pooled in a 15 mL tube per experimental condition and incubated with 200 μL of Accutase for 10 min. After homogenization and dilution with complete medium, cell suspension was sorted by BD FACSAria^TM^ III flow cytometer, isolating red-fluorescent-positive cells (lentiviral infected cells). Cells and isolated nuclei were sorted by flow cytometry and analyzed using FACs Diva Software (BD biosciences) to determine cell size and nuclei size. Red-fluorescent-positive cells were seeded at a density of 2.5 cells/μL in a 96-well ultralow-attachment plate (PrimeSurface^®^ 3D culture S-Bio) to obtain a single pure isolated clone per well. Cells were incubated at 37 °C with 5% CO_2_ for 7 days in complete medium with growth factors (bFGF and EGF) added every 3 days. After 7 DIV, wells were photographed using a camera system coupled to an inverted microscope, and the primary sphere diameter of the single clonal neurospheres was estimated using Fiji analysis software and GraphPad Prism7^TM^. To estimate the average size of the neurosphere population, 120 neurospheres from a minimum of three independent cultures were measured, taking the mean of the population diameter as a measure of culture proliferation [12].

### 2.3. Lentiviral Production and Infection of NSC Cultures

Lentiviral constructs of interest were *LV-CRE-mCherry* (Addgene #27546), to express Cre recombinase and ablate CBP, p300, or both proteins, and *LV-RFP* (Addgene #17619) as a control. Lentiviral particles (LV) were produced as previously described (Lois et al., 2002). HEK293T (ATCC CRL-3216) cells were cultured in Dulbecco’s modified Eagle’s medium (DMEM) containing low glucose, 10% fetal bovine serum, nonessential amino acids, 100 U/mL penicillin, and 100 µg/mL streptomycin, and they were regularly tested for mycoplasma contamination. For lentiviral production, HEK293T cells were seeded in 25 cm dishes at a density of 10^7^ cells/dish, incubated overnight at 37 °C with 5% CO_2_, and transfected using the calcium phosphate (CaPO_4_) method with 20 mg of lentiviral expression vector (*LV-CRE-mCherry* or *LV-RFP*), 15 mg of *pCMV-d8.9* plasmid expressing the gag and pol viral genes, and 10 mg of *pCAG-VSVg*, a plasmid encoding the VSV-G envelope gene. At 48 h after transfection, supernatants were collected, centrifuged at 2000 rpm for 5 min, and filtered to eliminate cell debris. Viral particles were concentrated with ultracentrifugation at 25,000 rpm for 90 min and aliquoted for immediate use or long-term storage at 80 °C. Viral titers were determined by serial dilution on HEK293T cells and FACS analysis (BD FACSAriaä III Flow Cytometer), using noninfected cells as a negative control. Usually, with this protocol, viral titers were on the order of 10^8^–10^9^ viral particles/mL. Neural stem cell (NSC) cultures were transduced at DIV1 with concentrated lentivirus at a multiplicity of infection (MOI) of 1–10. To ablate CBP, p300, or both proteins, we infected NSC cultures with lentivirus to drive the expression of Cre recombinase with mCherry reporter (*LV-CRE-mCherry*) or with lentivirus with RFP reporter (*LV-RFP*) in control cultures. After 3–4 days in culture, NSCs had proliferated to form neurospheres, and lentivirus began to be expressed. Fresh NSC complete medium with growth factors was added every 2–3 days. 

### 2.4. Clonal Differentiation of Neurospheres

To assess differentiation potential and multipotency, DIV7 clonal neurospheres were plated onto 12 mm glass coverslips (VWR) coated with Matrigel^®^ (Corning^®^) in differentiation medium I (NSC complete medium supplemented with 10 ng/mL of bFGF) at 37 °C with 5% CO_2_ for 48 h to allow neurospheres to attach to the coated surface and produce neuronal and oligodendroglial progenitors. After 2 days, the differentiation medium was replaced with differentiation medium II (NSC complete medium supplemented with 2% FBS in the absence of mitogenic factors) at 37 °C with 5% CO_2_ for 13 days leading to the generation of a dense layer of astrocytes with neurons and oligodendrocytes. The medium was not changed for the remainder of the culture. Differentiated cultures at 15 DIV were used for single-nucleus RNA Sequencing or were fixed for immunocytochemistry.

To visualize migration of neurosphere cells across the surface in real time, clonal neurospheres were plated on a 24-well plate (Falcon) coated with Matrigel^®^ (Corning^®^) in differentiation medium I. Then, 24-well plates were inserted into IncuCyte FLR (Essen Biosciences) for real-time imaging, with whole wells imaged under 4× magnification every 30 min for a total of 48 h. Data were analyzed using the IncuCyte Confluence version 1.5 software. All IncuCyte experiments were performed in triplicate.

### 2.5. Immunocytochemistry and Immunohistochemistry

To analyze the expression of neural progenitor markers, pure clonal neurospheres at DIV7 in a 96-well plate were fixed with prewarmed 4% paraformaldehyde (PFA) prepared in phosphate-buffered saline (PBS, 0.01 M) for 15 min at room temperature followed by three washes with PBS (5 min each). Then, neurospheres were permeabilized with 0.25% Triton^TM^ X-100 (Sigma) in PBS for 10 min and incubated with blocking solution (1% BSA and 0.1% Triton^TM^ X-100 in PBS) for 1 h at room temperature. After blocking, they were incubated overnight at 4 °C with the following primary antibodies: rabbit anti-RFP, mouse anti-Nestin, rabbit anti-Nanog, rabbit anti-Ki67, rabbit anti-caspase, mouse anti-phospho-histone H3. Neurospheres were then rinsed three times with PBS (5 min each), and the Alexa Fluor™ secondary antibody incubation was subsequently performed at room temperature for 1 h. After washing three times with PBS (5 min each), all neurospheres were incubated in 2 mM 4′,6-diamidino-2-phenylindole (DAPI, Sigma) nuclear stain for 10 min at room temperature. Neurospheres were transferred using a Pasteur pipette to coverslips for mounting in Mowiol^®^. To evaluate the multipotency of NSC cultures, coverslips with differentiated neurospheres at DIV15 were fixed. The procedure was identical to that described above except that the primary antibodies used were to detect different neural cell types: anti-glial fibrillary acidic protein (anti-GFAP) and anti-S100 calcium-binding protein B (anti-S100β) to detect astrocytes and mature astrocytes, respectively, as well as anti-beta III tubulin (TUBB3) to detect neurons and oligodendrocyte marker O4 (anti-O4) for oligodendrocytes.

### 2.6. Image Analysis and Quantification

Images were acquired using a super-resolution inverted confocal microscope Zeiss LSM 880-Airyscan Elyra PS.1 and Leica DM4000 microscope. Stained neurospheres and differentiated cultures were examined and photographed by super-resolution fluorescent microscopy (Zeiss). For quantitative analysis in differentiated neurospheres, images were acquired with a 25× objective, and cell quantification was performed in four regions of interest (ROIs) per differentiated neurosphere (a minimum of three coverslips per condition in three independent experiments) with image analysis software Fiji.

### 2.7. Single-Nucleus RNA Sequencing (snRNA-Seq) and Analysis

For the single-nucleus RNA-Seq experiment, a pool of ~15 differentiated neurosphere cultures from control, CBP^f/f^ + Cre, and p300^f/f^ + Cre at 15 DIV were used. Differentiation medium II was removed, and wells were washed with cold HBSS. Cold lysis buffer (10 mM Tris-HCl, 10 mM NaCl, 3 mM MgCl_2_, and 0.1% NonidetTM P40 Substitute in nuclease-free water) was added (500 µL) immediately to every well, gently pipetted until cells were completely suspended, and kept on ice for 5 min to lyse cells. The nuclear suspension from the different samples (control, CBP^f/f^ + Cre, and P300^f/f^ + Cre) was transferred to a 15 mL tube and centrifuged at 500 rcf for 5 min at 4 °C. The supernatant was removed, and the nuclear pellet was resuspended in 1 mL of Nucleus Wash and Resuspension Buffer (1× PBS with 1.0% BSA and 0.2 U/μL RNase Inhibitor). Centrifugation and nucleus wash steps were repeated twice. The nuclear suspension was sorted by the expression of RFP/mCherry on FACS Aria III (BD Biosciences) at 4 °C and immediately proceed according to the 10x Genomics single-cell protocol. A total of 16,000 nuclei per sample (pool of ~15 differentiated neurosphere cultures) were loaded into single cell Chromium Chip B, and Chromium Controller was run to generate partitioning thousands of nuclei into nanoliter-scale gel beads in emulsion (GEMs) and immediately reverse-transcribed, where all generated full-length cDNA (from polyadenylated mRNA) shared a common 10× Barcode. Post-GEM-RT cleanup, cDNA amplification, and 3ʹ gene expression library construction were performed using a Chromium Single-Cell 3′ Library and Gel Bead Kit v3. Quality control of cDNA and libraries was performed using a Agilent Bioanalyzer in a high-sensitivity chip. Libraries were sequenced together in a unique cell flow on an Illumina HiSeq2500 sequencer to obtain 75 bp paired-end reads following the manufacturer’s instructions, with an average sequencing depth of 290 million reads per sample. Quality control was performed using FastQC v0.11.9. Sequenced samples were processed using the Cell Ranger v3.1 pipeline (10× Genomics) and aligned to the GRCm38 (mm10) mouse-reference genome customized to count reads in exons and introns (pre-mRNA) (gene annotation version 94). Bioinformatic analyses were performed as described previously [13,14]. Single-nucleus RNA-seq data were preprocessed and further analyzed in R using Seurat v2.3.4 [15,16] with the following filter parameters: genes, nCell < 5; cells, nGene < 200. Barcodes with a total unique molecular identifier (UMI) count >10% of the 99th percentile of the expected recovered cells were selected for further analysis. We retrieved 4884 (control), 3017 (*Crebbp^f/f^* + Cre), and 4648 (*Ep300^f/f^* + Cre) high-quality nuclei per sample. Mean reads per nucleus were 62,810 (control), 99,066 (*Crebbp^f/f^* + Cre), and 65,928 (*Ep300^f/f^* + Cre). Data were then normalized using global scaling normalization (method: LogNormalize, scale.factor = 10,000). To identify major cell populations in the differentiated cultures, datasets were analyzed separately.

Highly variable genes (HVGs) were detected using the *FindVariableGenes* function with default parameters. Then, normalized counts on HVGs were scaled and centered using the *ScaleData* function with default parameters. Principal component analysis (PCA) was performed over the first ranked 1000 HVGs. Plots of the two principal components of the PCA, where cells were colored by dataset, excluded the presence of batch effects. Cluster detection was carried out using the Louvain algorithm with the first 20 PCA dimensions at resolution = 0.6 (the default and the optimal according to cell number, data dispersion, and co-expression of previously reported cell markers). Visualization and embedding were performed using t-distributed stochastic neighbor embedding (tSNE) [17] and uniform manifold approximation and projection (UMAP) [18] over PCA using the first 20 PCA dimensions.

After characterizing the cell heterogeneity of each condition separately, data from the control, CBP-ablated, and p300-ablated conditions were merged into a single dataset. Using the first 1000 HVGs from each sample, a total of 1421 unique HVGs detected in the three conditions were obtained. Normalized counts on merged HVGs were scaled and centered using the *ScaleData* function with default parameters, and PCA was performed over them. Plots of the two principal components of the PCA, where cells were colored by dataset, excluded the presence of batch effects. Subsequent analyses repeated the same strategy and the same clustering and visualization parameters as in the individual analyses. The *FindAllMarkers* function with default parameters was used to identify gene markers for each cluster and to assign the cell-type identity. Populations were identified by combining the clustering results over the merged dataset with clustering information obtained from the datasets separately, together with the co-expression of population markers and information from external single-cell RNAseq (scRNA-seq) datasets from the Allen Brain Map portal [19] and the Atlas of the Developing Mouse Brain [20]. Twelve clusters were unbiasedly identified, but three of them were closely distributed and manually fused since they presented minor expression differences. UMAP plots of gene expression show the normalized count (UMIs) per nucleus. Clustering was performed on the merged dataset from control, CBP-ablated, and p300-ablated conditions, and populations were identified by combining these results with clustering information obtained in the datasets separately, together with the co-expression of population markers. The *FindAllMarkers* function with default parameters was used to identify gene markers for each cluster and to assign cell-type identity to clusters based on scRNA-seq dataset of the Allen Brain Map portal (Mouse Whole Cortex and Hippocampus SMART-seq (2019) with 10x SMART-seq taxonomy (2020).

Differential expression analysis (DEA) was used to identify population gene effects between the different conditions. For DEA, the nuclei of each population were compared against all the other nuclei in the merged dataset using the Wilcoxon rank sum test on normalized counts with the following parameters: logfc.threshold = 0.25; min.pct = 0.25. Gene Ontology (GO) functional enrichment analyses for biological process (BP) were performed using clusterProfiler (v3.15.2). All enriched terms were considered significant at adjusted *p*-values by BH cutoff < 0.05. The reference gene set used to perform the analysis was the C5 (GO Biological Process) collection from the Molecular Signatures Database (MSigDB) (v6.2).

### 2.8. Statistical Analyses

Statistical analyses were carried out in GraphPad Prism8 Software. Sample size was based on previous experience and the literature. Data were presented as the mean ± standard error of mean (SEM). Statistical comparison between two groups was first tested for normality using Kolmogorov–Smirnov or Shapiro–Wilk tests and for variance with F-test. Data with a normal distribution were compared using the unpaired two-tailed Student’s t-test. When the normality test failed, the Mann–Whitney U-Test was performed. Statistical significance was defined for *p*-values below 0.05. * *p* < 0.05, **: *p* < 0.01, and *** *p* < 0.001.

## 3. Results

### 3.1. NCS Proliferation Is Preserved in the Absence of CBP or p300 but Impaired When Neither KAT3 Protein Is Present

To investigate CBP functioning during development we used a previously reported *Crebbp* conditional knockout (cKO) line in which exon 7 (encoding part of the KAT domain) is flanked by loxP sites (*Crebbp*^f/f^) [21] (Figure 1a). After recombination, the locus expresses a truncated CBP protein that lacks the KAT domain (aa 1300–1700) and other critical domains (KIX and CH3) downstream of the recombination site. To study the role of p300 in development, we also used a previously reported conditional *Ep300* KO line [7] in which exon 9 (encoding part of the KIX domain) is flanked by loxP sites (*Ep300*^f/f^; Figure 1a). After cre-dependent recombination, this genetic manipulation results in the absence of mature mRNA and protein, as confirmed by qRT-PCR, immunohistochemistry, and Western blot [7]. The removal of both KAT3 proteins simultaneously can be achieved in mice carrying homozygous floxed alleles for CBP and p300 (*Crebbp*^f/f^x*Ep300*^f/f^; Figure 1a).

To generate neurospheres lacking CBP, p300, or both, we isolated NSCs from the SVZ of *Crebbp*^f/f^, *Ep300*^f/f^, and *Crebbp*^f/f^x*Ep300*^f/f^ newborn mice (P0–P3) and infected them with a lentiviral vector encoding either the red fluorescent protein (RFP), as a control, or a Cre recombinase fused to mCherry to target the floxed alleles. Cells were then seeded in growth medium supplemented with epidermal growth factor (EGF) and fibroblast growth factor (FGF) as mitogens in the absence of adherent substrate (Figure 1b). After 2 weeks in culture to allow complete removal of KAT3 proteins, primary neurospheres were isolated by fluorescent-activated cell sorting (FACS) and replated to generate clonal secondary neurospheres. Next, we measured neurosphere diameter as an approximation to quantify cell proliferation [22]. CBP-lacking neurospheres showed similar diameters to control neurospheres after 7 days in culture post sorting (DIV7) (Figure 1c,d), suggesting that CBP does not interfere with proliferation. However, in the case of p300, we observed an increase in the diameter of the neurospheres (Figure 1e,f). To test whether this increase in diameter was a consequence of a larger cell size or a higher proliferation rate, we quantified the number and size of p300-ablated NSCs by FACS. We did not find significant differences in cell number (19,397 ± 637.5 vs. 19,453 ± 522.5, *p* = 0.9488). However, NSCs lacking p300 presented a slightly larger cell size (137.193 ± 36.646 *vs.* 64.744 ± 16.959, *p* < 0.0001) and nucleus size (mean 73.82 ± 14.677 *vs.* 69.84 ± 6.454, *p* < 0.0001) compared to controls (Figure 1g,h). This result suggests that p300 elimination impacts cell size but does not affect the proliferation of NSCs. Importantly, neurospheres with both CBP and p300 depleted showed a great reduction in diameter compared to control neurospheres (Figure 1i,j). Overall, these results indicate that, whereas proliferation of NSC is not affected by the absence of CBP or p300, simultaneous removal of both proteins seriously compromises proliferation and self-maintenance, indicating that the presence of at least one of the two KAT3 proteins is required for NSC proliferation.

### 3.2. CBP and P300 Are Individually Required for Proper NSC Differentiation

Neurospheres plated onto an adherent substrate in the presence of serum induced differentiation after 2 weeks in a typical profile: 80% of the cells in the culture expressed the glial fibrillary acidic protein (GFAP) (presumably astrocytes), 17% expressed neurofilament (NF) (presumably neurons), and 1–3% expressed O4 (presumably oligodendrocytes) (Reynolds and Weiss, 1992) [23,24]. We took advantage of this assay to investigate the capacity of neurospheres lacking CBP or p300 to produce different neural lineages. The 7DIV clonal secondary neurospheres were plated in the absence of the mitogen EGF to allow neurospheres to attach to the coated substrate and differentiate (Figure 2a); 2 days later, the medium was supplemented with serum, and the removal of CBP and p300 was confirmed (Appendix A). Then, we analyzed the migration and differentiation of cells exiting the neurospheres by performing time-lapse recordings. As early as 48 h after inducing differentiation, we already observed differences between CBP-KO and p300-KO, as well as their respective control cultures. CBP-lacking cultures exhibited a strong delay in cell migration compared to control cultures, whereas p300-ablated cells showed slightly slower migration than control cells (Figure 2b,c and Appendix A). Neurospheres mutant for both KAT3 proteins did not self-replicate (Figure 1i,j); consequently, the differentiation assay was not possible to perform with the double-mutant NSCs.

After maintaining the cultures of CBP-KO, p300-KO, and control neurospheres for 15 DIV, we performed immunostaining with TUBB3, to label differentiating neurons, and GFAP, which is a stemness marker at early stages of differentiation and labels astrocytes at later stages. Control cultures stained with GFAP exhibited typical stellate with large filamentous morphology while a non-ramified morphology and low level of GFAP immunoreactivity were observed in neurospheres lacking CBP or p300 (Figure 3a–c’) (0.78 fluorescence intensity (arbitrary units) ± 0.018 (SEM)/nuclei (DAPI) in the control vs. 0.017 ± 0.035/nuclei in CBP-ablated and 0.01 ± 0.022/nuclei in P300-ablated samples; *p* < 0.0001). Furthermore, in the CBP-KO and p300-KO cultures, there were very few TUBB3^+^ cells, suggesting that neural progenitors lacking CBP or p300 do not differentiate properly into neuronal fate (Figure 3d–f, Figure 4a). We also stained with markers for mature astrocytes (S100ß) and oligodendrocytes (O4) and found cells positive for each of these markers in the control cultures in the expected proportions [23,24]. However, there was a significant loss of TUBB3^+^ cells in CBP- and in p300-lacking cultures compared to the controls (Figure 4a–d) and a dramatic decrease in the proportion of S100ß^+^ cells in both CBP and p300 mutant cultures (Figure 4e–g). As in the controls, cultures lacking CBP or p300 showed fewer than five O4^+^ cells per plate (Figure 4i–k’). We also measured cell death after the elimination of CBP and p300 and observed no significant changes in the number of dying cells in either of the two genotypes compared to the controls (Figure 4l–o).

These results suggested that NSCs lacking either CBP or p300 can survive in vitro but are not capable of differentiating properly into the different neural lineages.

### 3.3. Single-Nucleus RNA Sequencing Analysis of Neurospheres-Derived Cultures Revealed Differential Alterations after Removal of CBP or P300

To explore the molecular changes underlying the morphological alterations observed in the CBP- and p300-deficient cultures, we conducted single-nucleus RNA sequencing (snRNA-seq) in the differentiation-induced cultures. After quality control (Appendix A), nucleus filtering, and joint analysis of samples from the two conditions and the controls, we performed unsupervised consensus clustering to identify major cell types and canonical marker genes (Appendix A). The integrated bioinformatic analysis identified 12 clusters (Figure 5a) that were annotated to 10 cell types according to the top 20 bona fide marker genes per cluster from the scRNA-seq dataset of the Allen Brain Map portal [19] and markers from the Atlas of the Developing Mouse Brain [20] (Figure 5b). Gene expression in these clusters identified cluster 2 as glial/astrocyte progenitor cells (GPC) (*Mgat5* and *Aqp4*), cluster 3 as astrocytes (*Slc1a3, Gpc5,* and *Atp1a2*), cluster 4 as oligodendrocyte progenitor cells (OPC) (*Bcas1, Sirt2,* and *Enpp6*), cluster 5 as immature neurons (INMN) (*Kcnj6, Nell2,* and *Cacng4*), cluster 6 as oligodendrocytes (*Mog, Mbp,* and *Mag*), cluster 7 as proliferating cells (*Mki67, Kif4,* and *Rbfox1*), cluster 8 as neurons (*Ascl2, Cpn4,* and *Adarb2*), and cluster 9 as microglia (*Inpp5d, Fyb,* and *Fli1*). The remaining clusters were not characterized by a single cell type, suggesting a spectrum of cells in transition between different states; cluster 0 appeared to be a mix of neuronal progenitors (NPCs) and OPCs, while cluster 1 appeared to be a mix of pluripotent stem cells and/or radial glial cells (PSCs/RGCs).

Uniform manifold approximation and projection (UMAP) plots displaying differential clustering showed a complete absence of the astrocytic population (cluster 3) both in CBP-lacking and in p300-lacking cultures (Figure 5c). This cluster included bona fide marker genes for astrocytes, such as *Scl1a3* (GLAST*), Scl7a11, Gpc5, Apoe, Rorb, Atp1a2*, and *Ntm*. Next, we explored whether the absence of astrocytes was compensated for by alterations in other populations. Analysis of cluster proportion revealed that several clusters displayed altered nuclei proportions in CBP-lacking and/or p300-lacking cultures compared to the controls (Figure 5d). Cluster 0 (NPC/OPCs) showed a lower proportion of cells in the CBP-ablated cultures than in the control, while the p300-lacking cultures were slightly more populated than the controls. Cluster 5 (INMN) showed no alteration in terms of proportions but exhibited differences in cell distribution in CBP-deficient cultures compared to the control or the p300-ablated conditions (Figure 5c,d). Together, these results indicate that (i) both KAT3 proteins are essential for astrocytic lineage differentiation, (ii) the absence of CBP strongly interferes with NPC/OPCs differentiation favoring the oligodendrocytic lineage and with INMN maturation, and (iii) the absence of p300 leads to defects different to those observed after CBP ablation.

Next, we investigated the transcriptional alterations associated with CBP removal in the two clusters that displayed the strongest effect (cluster 0 and cluster 5) (Figure 6a; Appendix A). In the CBP-lacking cultures, 44 genes were upregulated and 66 were downregulated in cluster 0, while 112 were upregulated and 205 were downregulated in cluster 5. To identify biological processes affected by these transcriptional changes, we performed GO overrepresentation analysis. Downregulated DEGs in cluster 0 showed robust association with GO-BP terms (adj. *p* < 0.05) linked to neuronal functions, such as *NMDA glutamate receptor clustering*, *neuron cell–cell adhesion*, *presynaptic membrane organization*, *glutamatergic synaptic transmission*, and *synapse assembly*, including many well-known modulators of these processes such as *Nrxn1, Grik2, Sema3d, Nlgn1*, and *Ntm*. In contrast, the analysis of the upregulated genes (e.g., *Mbp, Plp1, Tcf7l2, Bcas1, Sox6*, and *Sox10)* retrieved terms related to oligodendrocyte fate such as *oligodendrocyte development*, *ensheathment of neurons*, *axon ensheathment*, or *myelination* (Figure 6a,b). GO analysis for cluster 5 also evidenced a clear trend for this population to adopt an oligodendrocytic lineage to the detriment of neuronal markers because there was a highly significant enrichment of upregulated genes in *myelination*, *ensheathment of neurons*, or *axon ensheathment* terms and an enrichment in terms such as *post-synapse organization*, *regulation of synapse organization*, *modulation of chemical synaptic transmission*, *chemical synaptic transmission*, or *neuron development* among the downregulated DEG (Figure 6c,d). In contrast to the CBP-lacking cultures, the upregulation of genes associated with the oligodendrocytic lineage was not evident in any of these two clusters in the p300-ablated cultures. Together, these results confirm the key role for CBP in the regulation of gene programs implicated in neuronal/oligodendrocyte lineage decision and the different function for p300 in the differentiation of NSC.

Altogether, immunostaining data and differential gene expression analyses based on snRNA-seq confirmed profound transcriptional changes that support important individual roles for both KAT3 proteins in astrocytic development. In addition to this essential role in astrocytes, our results are compatible with the idea that CBP and p300 play nonredundant roles in the development of other neural types. While elimination of each of these proteins induces arrest of neural progenitors, only the absence of CBP triggered aberrant differentiation to the oligodendrocyte lineage (Figure 6e).

## 4. Discussion

Conventional CBP and p300 KO mice are embryonic lethal [4,5] due to a failure in neural tube closure and extensive brain hemorrhage. This severe phenotype initially limited the precise investigation of the effects of CBP and p300 ablation in the CNS. These initial studies suggested that both KAT3 proteins play similar role in embryonic development because of the almost identical phenotype of conventional knockouts [4] and their sequence homology [25]. However, more recent studies have shown that, in different tissues, the functions of these proteins do not always overlap. For instance, p300 is required for the development of lung or heart [26,27], whereas CBP has a prominent role in motor neuron differentiation [28]. Given these observations, we decided to compare the function of KAT3 in proliferation and differentiation of NSC by generating neurospheres from the SVZ and differentiating them to the different neural lineages [6].

Previous work by Wang and collaborators [29] showed that CBP knockdown in vitro had no effect on the percentage of proliferating cortical precursor cells expressing Ki67 or cleaved caspase-3, markers of proliferation and apoptosis, respectively, consistent with our neurospheres and RNA-seq analysis. However, our experiments went further and demonstrated that, although proliferation of neural progenitor cells is not affected in the absence of CBP nor p300, it is seriously compromised when both proteins are absent.

In addition, neurosphere-induced differentiation showed that elimination of either CBP or p300 interfered with glial and neuronal differentiation. Very few S100ß^+^ cells were observed in CBP- or p300-lacking cultures, and the morphology of GFAP-positive cells was also altered in both conditions compared to the control differentiated cultures. Our snRNA-seq assay helped to precisely determine the molecular basis of the dramatic change in the morphology of GFAP and reduction in the number of S100ß^+^ cells. Even further, the bioinformatic analysis of these cultures revealed that the astrocyte population completely disappeared after ablation of either of the two KAT3 proteins. The cluster population expressing bona fide markers that define astrocytes, such as *Scl1a3* (GLAST), *Scl7a11*, *Gpc5*, and *Rorb*, completely disappeared in CBP- or p300-lacking cultures. The disappearance of cluster 3 was accompanied by an increase in other populations such as cluster 4 (presumably OPCs) and cluster 6 (presumably oligodendrocytes) in the case of CBP-lacking cultures or cluster 0 (NPCs/OPCs) in the case of p300-lacking cultures. In line with these observations, OPC- and oligodendrocyte-specific genes were significantly upregulated in the absence of CBP [23,24]. These results suggest that, in the absence of CBP, cells that should have acquired an astrocytic fate might have acquired an oligodendrocyte lineage instead. However, when p300 was eliminated, cells that could not differentiate to astrocytes were stalled at more progenitor/pluripotent stages.

According to GO enrichment analyses, in a large population of immature neurons in CBP- or p300-lacking cultures, many genes involved in neuronal differentiation, cell polarity and adhesive functions, axonal projection and guidance, or synapse organization and assembly, including well-known modulators of these processes such as *Nrxn1*, *Grik2*, *Sema3d*, *Nlgn1*, and *Ntm*, were downregulated. In line with these results, it has been shown that iPSC-derived neurons from RSTS patients have an altered neuroprogenitor-to-neuron transcriptional program [30].

We also noticed in our snRNA-seq analysis that cluster 8, which according to the expression of markers such a NeuN or Dcx may be identified as a neuronal population, was not altered in the absence of CBP or p300. Neurospheres contain a heterogeneous population of cells in which NSCs coexist with their progeny, committed progenitors, and even differentiated cells [31,32]. Because it has been recently demonstrated that elimination of either one of the KAT3 proteins does not affect the maintenance of neuronal fate [13], it is possible that the small subset of differentiated neurons integrating the neurosphere may have lost CBP or p300 after acquiring their final fate. Supporting this, cluster 8 in our snRNA-seq UMAP analysis likely corresponds to the population of neurons that were already differentiated in the free-floating neurosphere. Another possibility is that cluster 8 represent cells derived from defective cre-dependent recombination and, thus, were able to differentiate.

Overall, our work demonstrates that NSCs lacking CBP or p300 are not able to differentiate properly to lead to the three main neural lineages, and we identified the specific transcriptional changes that accompanied these alterations. The results also indicate that the astrocyte fate is very sensitive to the ablation of either of the two KAT3 proteins, while the mature neuronal phenotype could not be achieved in CBP-ablated NSCs, giving rise instead to differentiation of the oligodendrocyte lineage. By contrast, p300-ablated NSCs remained in pluripotent stages showing mixed lineages, further demonstrating the different functions of these two related co-activators/acetyltransferases in the brain. These findings shed new light onto the etiology of RSTS and unveil new aspects of the role of KAT3 proteins in neural development.

## Figures and Tables

**Figure 1 cells-11-04118-f001:**
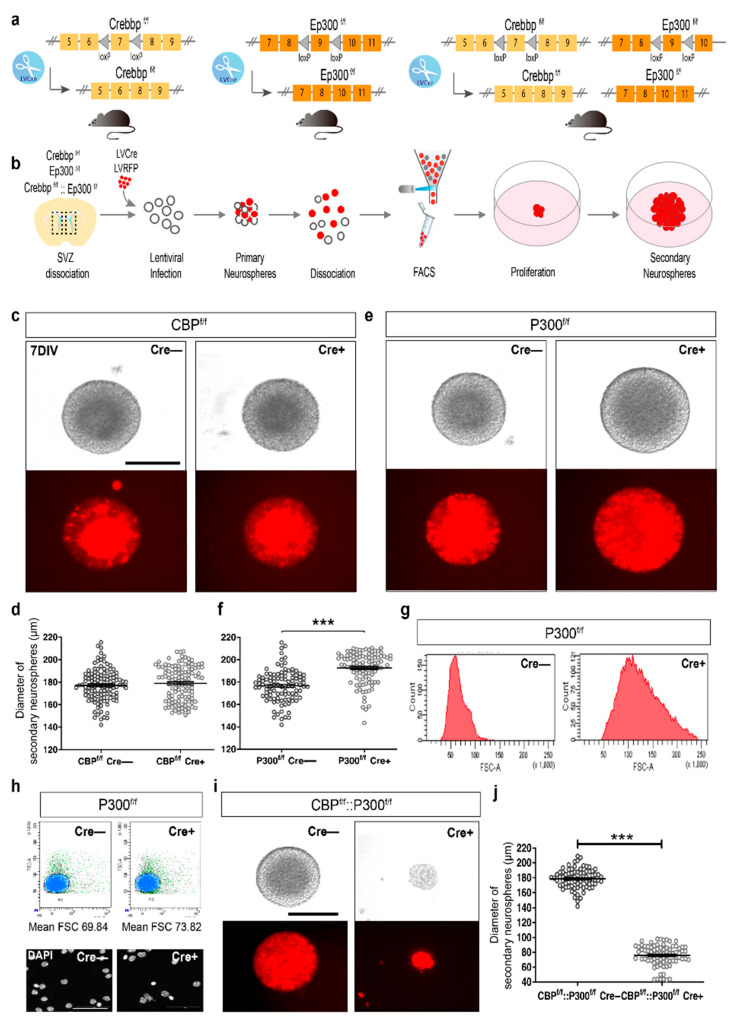
Individual ablation of CBP or p300 does not affect proliferation of NSC. (**a**) Scheme representing the conditional lines generated to study the ablation of KAT3 proteins in different developmental processes. (**b**) Experimental design of the approach used to generate neurosphere cultures to investigate the role of KAT3 proteins in proliferation. (**c**) Representative clonal secondary neurosphere lacking CBP at 7 DIV. (**d**) Quantitative analysis of the diameter of CBP-lacking clonal secondary neurospheres at 7 DIV (*n* ≈ 120 neurospheres from three independent experiments). (**e**) Representative clonal secondary neurospheres lacking p300 at 7 DIV. (**f**) Quantitative analysis of the diameter of clonal secondary neurospheres lacking p300 at 7 DIV; *** *p* < 0.001 (*n* ≈ 120 neurospheres from 3 independent experiments). (**g**) Flow cytometry sorting plots for infected cells from controls and p300-ablated NSCs showing counts against forward scatter (FSC). (**h**) Fluorescent-activated sorting analysis plots showing forward scatter (FSC) against red fluorescence of nuclei from controls and p300-ablated NSC (*n* = 3; each sample contains a pool of 15 neurospheres). The bottom panels show high magnification of nuclei counterstained against DAPI of control and p300-ablated NSCs. (**i**) Representative infected clonal secondary neurospheres lacking both KAT3 proteins at 7 DIV. (**j**) Quantitative analysis of the diameter of clonal secondary neurospheres lacking both proteins at 7 DIV *** *p* < 0.0001 (*n* ≈ 120 neurospheres from three independent experiments). Error bars denote the SEM. Scale bar: (**c**–**e**) 200 μm and (**i**) 50 μm.

**Figure 2 cells-11-04118-f002:**
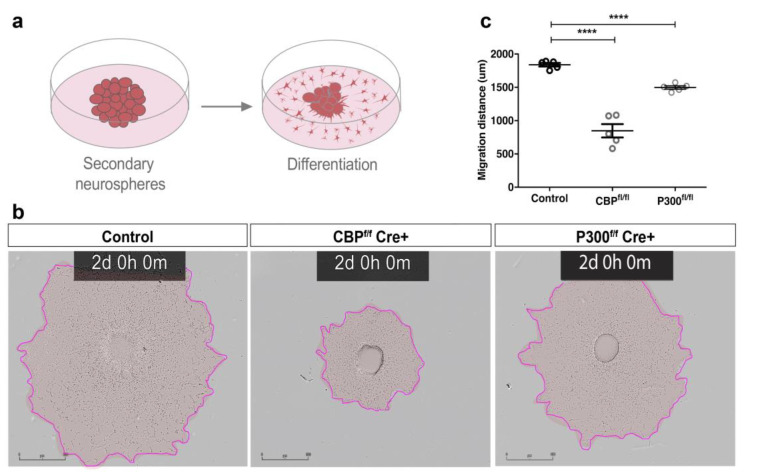
CBP-lacking neurospheres showed a prominent delay in cell differentiation and alterations in neural lineage markers. (**a**) Scheme representing differentiation of clonal secondary neurospheres. (**b**) Overlaid brightfield depicting cell migration obtained by real-time IncuCyte imaging at 48 h. (**c**) Quantitative analysis of cell migration (μm) after 48 h (*n* = 5 per genotype). CBP-lacking neurospheres showed a prominent delay in cell migration, whereas p300-lacking neurospheres exhibited a slightly reduction compared to control neurospheres. Scale bar: 800 µm. **** *p* < 0.0001.

**Figure 3 cells-11-04118-f003:**
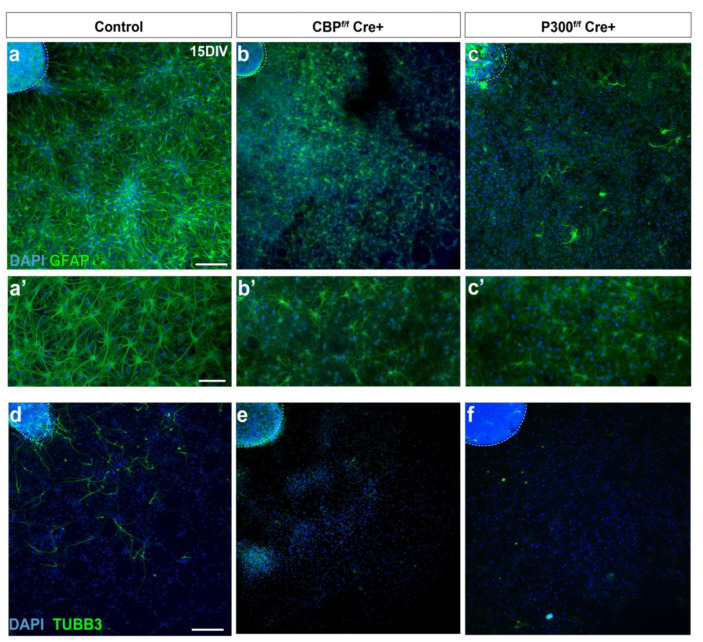
Differentiating neurospheres lacking either CBP or p300 show alterations in neural lineage markers. (**a**–**f**) Representative images of differentiating neurospheres lacking CBP or P300 at 15 DIV exhibit alterations in GFAP- and TUBB3-positive cells compared to the controls. (**a’**–**c’**) Higher magnification of the cultures stained with GFAP. Scale bars: 200 µm (**a**–**f**) and 100 µm (**a’**–**c’**).

**Figure 4 cells-11-04118-f004:**
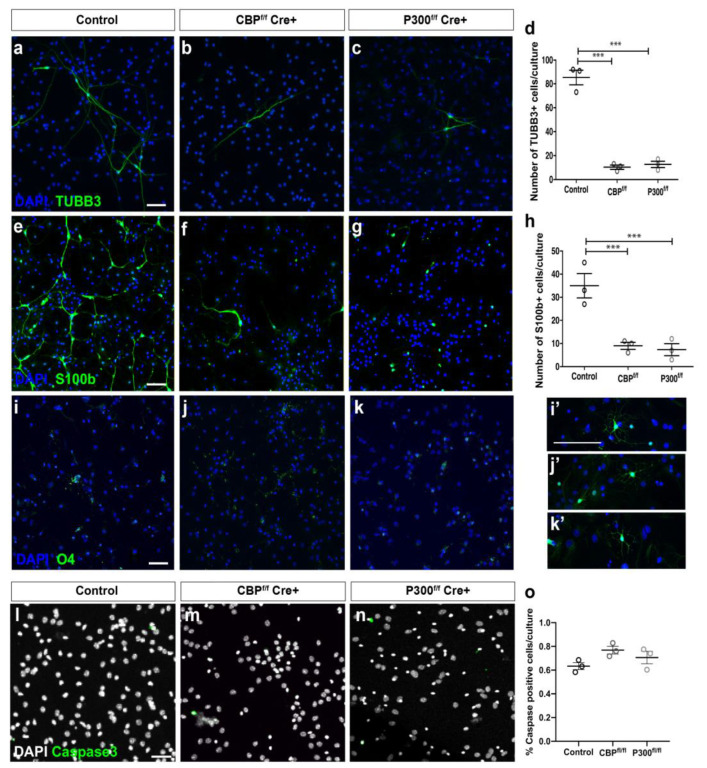
CBP and p300 are individually required for differentiation of NSCs into different neural lineages. (**a**–**k**) Representative images of immunofluorescent staining for TUBB3, S100ß, or O4 in differentiated CBP- or P300-ablated neurospheres at 15 DIV. (**d**,**h**) Quantification of neurons and mature astrocytes in neurospheres lacking CBP or P300 after 15 DIV of differentiation. (**i’**–**k’**) High-magnification examples of the few O4^+^ cells in control, CBP-ablated, and P300-ablated cultures; *** *p* < 0.001 (*n* = 5 coverslips per condition from three independent cultures) according to one-way ANOVA. Error bars denote the SEM. Scale bars: 100 µm. (**l**–**n**) Immunostaining against caspase 3 and quantification of caspase 3-positive cells per culture showed no differences in cell death after differentiation of NSCs lacking CBP or p300 compared to control NSCs. Scale bars: 100 µm. (**o**) Quantification of the number of caspase 3 cells (*n* = 3 coverslips per condition from three independent cultures) according to one-way ANOVA. Error bars denote the SEM.

**Figure 5 cells-11-04118-f005:**
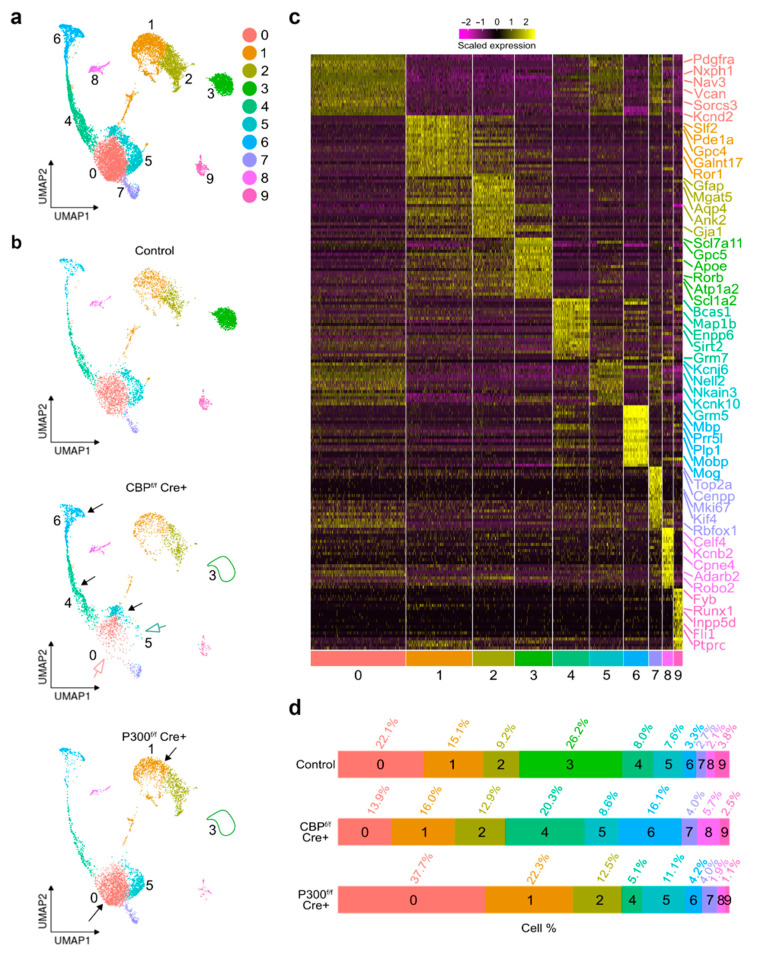
snRNA-seq analysis of differentiating neurospheres reveals that CBP and p300 are both required for astrocyte differentiation but play different roles in the differentiation of the other neural lineages. (**a**) UMAP plots displaying differential clustering on differentiated neurospheres at 15 DIV of control, CBP-ablated, and p300-ablated nuclei combined. Dots represent single nuclei. Nuclei are colored by their classification label. (**b**) UMAP embedding of single-nucleus profile for control, CBP-lacking, or p300-lacking cultures. Note that cluster 3 disappears in CBP- and p300-lacking cultures. Empty arrows point to a reduction in the number of cells, while black arrows highlight an increase in cell number compared to the respective control clusters. (**c**) Heatmap of z-score of single-nucleus expression levels of the five top-ranked bona fide marker genes, with selected genes displayed on the *y*-axis and cells displayed on the *x*-axis. Each row represents a gene, the columns are nuclei, and the color code represents the normalized expression for upregulated genes (yellow) or downregulated genes (purple). (**d**) Bar plot showing the percentage of nuclei in each cluster of cultures from the three different genotypes.

**Figure 6 cells-11-04118-f006:**
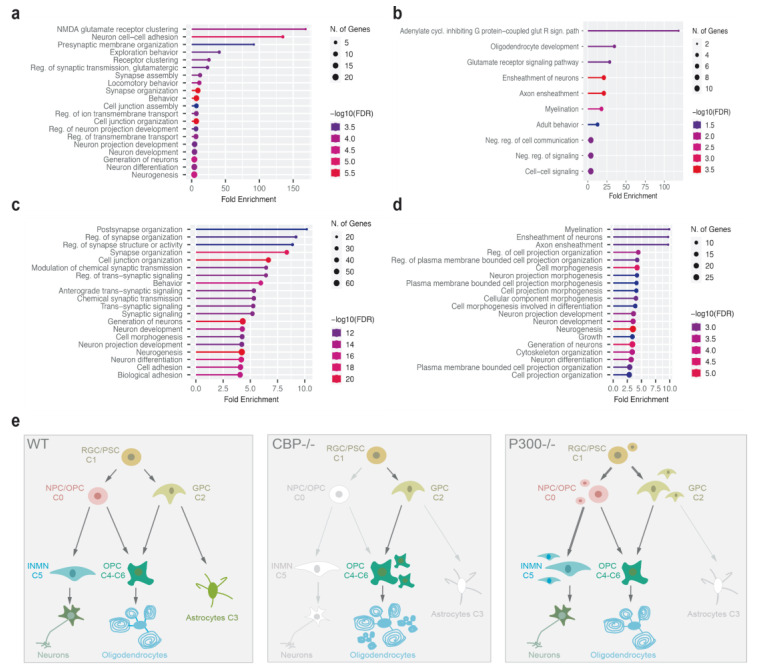
Gene programs altered during neural differentiation in the absence of CBP or p300. (**a**,**b**) Visualization of Gene Ontology (GO) analysis for downregulated (**a**) and upregulated (**b**) genes in cluster 0 in CBP-lacking cultures. (**c**,**d**) Visualization of Gene Ontology (GO) analysis for downregulated (**c**) and upregulated (**d**) genes in cluster 5 in CBP-lacking cultures. (**e**) Working model for CBP and P300 functions in neural differentiation. NSCs are tripotent cells. During their differentiation, they give rise to transiently dividing progenitors (transit amplifying progenitors) that subsequently undergo lineage restrictions toward neuronal, astrocytic, and oligodendroglial mature cells. NSCs lacking CBP or p300 are not able to differentiate properly to lead to the three main neural lineages. The astrocyte fate is particularly sensitive to the individual lack of KAT3 proteins, and the presence of one of them does not compensate for the absence of the other. CBP elimination leads to a tendency toward oligodendrocytic lineage differentiation. In contrast, in the absence of p300, NSCs are not able to reprogram in a different lineage, rather stalling at different pluripotent stages. RGC, radial glial cells; PSC, pluripotent stem cells; OPC, oligodendrocytes progenitor cells; GPC, glial progenitor cells.

## Data Availability

The snRNA-seq data discussed in this publication were deposited in NCBI’s Gene Expression Omnibus [33] and are accessible through GEO Series accession number GSE211737 (https://www.ncbi.nlm.nih.gov/geo/query/acc.cgi?acc=GSE211737).

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
