# Peer review of "CBP and p300 Jointly Maintain Neural Progenitor Viability but Play Unique Roles in the Differentiation of Neural Lineages"

_cells, 2022, doi:10.3390/cells11244118_

Round 1
Reviewer 1 Report
The manuscript presented by Herrera and collaborators aims to describe the role of two paralogous lysine acetyltransferase proteins, CBP and p300, in the proliferation and differentiation capacity of Neural Stem Cells derived from the Subventricular zone in P0-P3 mouse pups. The manuscript presents a proper experimental design, a detailed explanation of methods and results, and adequately supported conclusions. The manuscript may be published in its current form but it would be ideal if the authors make the following changes:
In the methods section, Line 216, please clarify what reporter was used to sort the nuclei suspension.
In the results section, Line 324, please clarify how was quantified the nucleus size based on Forward scatter (FSC).
In Figure 3 is really difficult to appreciate the morphological changes described in the text (Lines 353, 354), please include pictures with a higher magnification. In addition, it will be ideal to present a quantification of roundness between groups, supporting the increase of cell proportion with a non-ramified morphology in the experimental groups.
In figure 4 the number of O4 positive cells seems to be higher than the number of S100b or TUBB3, however, the manuscript states that due to low numbers, a quantification of O4 positive cells was not possible, please elaborate on the reasons for this discrepancy.
Given the ambivalence of GFAP as an astrocytic marker, is it possible to use alternative markers to support via immunocytochemistry the absence of cluster 3 in CBP and p300 lacking groups described in Figure 5C.
Author Response
The manuscript presented by Herrera and collaborators aims to describe the role of two paralogous lysine acetyltransferase proteins, CBP and p300, in the proliferation and differentiation capacity of Neural Stem Cells derived from the Subventricular zone in P0-P3 mouse pups. The manuscript presents a proper experimental design, a detailed explanation of methods and results, and adequately supported conclusions. The manuscript may be published in its current form but it would be ideal if the authors make the following changes:
Authors answer (aa): We thank the Reviewer for the very positive appreciation of our work.
In the methods section, Line 216, please clarify what reporter was used to sort the nuclei suspension.
aa: We have added “nuclei were sorted by the expression of RFP/mCherry” (line 214)
In the results section, Line 324, please clarify how was quantified the nucleus size based on Forward scatter (FSC).
aa: Nuclei were isolated before sorting and we used the BD FACSDiva Software to determine particule size. This has been now stated in Material and Methods section (Line 119).
In Figure 3 is really difficult to appreciate the morphological changes described in the text (Lines 353, 354), please include pictures with a higher magnification. In addition, it will be ideal to present a quantification of roundness between groups, supporting the increase of cell proportion with a non-ramified morphology in the experimental groups.
aa: We have now replaced the images in Fig. 3 for better resolution pictures and it is possible to appreciate the morphological differences between cells from the different genotypes. We appreciate the suggestion of the referee about the quantification of roundness. However, this is very difficult to perform because the celular density and the blurness of the staining in the mutant cultures does not allow an easy visualization of individual cells.
In figure 4 the number of O4 positive cells seems to be higher than the number of S100b or TUBB3, however, the manuscript states that due to low numbers, a quantification of O4 positive cells was not possible, please elaborate on the reasons for this discrepancy.
aa: The number of O4 positive cells is actually very low; around 2-4 per culture. Perhaps the referee was confused because the pictures in panels Fig4. I-K were at higher resolution than the ones for the other markers. We are now providing lower magnification images of O4-labeled cultures to show that there are very few O4+ cells. We also show high magnification images of O4+ cells from each genotype just to demonstrate that the stainig worked. Because of this small number of O4+ cells, we would need a very large number of cultures to perform proper statistics. In fact, this is why we decided to perform the single cell experiments. We have now removed the sentence stating that it was not possible to quantify the low numbers of O4 cells, and replaced it by the following sentence: “As in the controls, cultures lacking CBP or p300 showed less than 5 O4+ cells per plate”.
Given the ambivalence of GFAP as an astrocytic marker, is it possible to use alternative markers to support via immunocytochemistry the absence of cluster 3 in CBP and p300 lacking groups described in Figure 5C.
aa: The reviewer is right. GFAP is actually an ambivalent marker and this is why we stained the cultures with S100ß which is a specific marker for astrocytes (Fig. 4E-G, H). S100 ß staining support the finding that there are no astrocytes in the CBP and p300 lacking cultures.
Reviewer 2 Report
In this paper, R Gonzalez-Martinez et al. investigated the role of the co-activator/acetyltransferases CBP/P300 (KAT3) in defining the cell lineage from progenitor brain cells. They took advantage of the production of conditional Crebbpf/f, Ep300f/f or Crebbpf/f x Ep300f/f mice to generate neural stem cell (NSC) cultures lacking either one or both proteins. Protein ablation was induced by viral infection driving cre recombinase expression and FACS isolation of specific clones that could be cultured in vitro. Lineage fate was evaluated using different parameters assessing proliferation, differentiation, cell-spedific protein expression (immunohistochemistry), and the cell-specific transcriptomics (snRNA-seq). This is a very nice work that, with a “quite simple” and elegant in vitro system, can thoroughly assess and demonstrate important functions for CBP and P300 in cellular fate decisions. Especially the snRNA-seq analyses further provided valuable information about the molecular composition of the produced lineages in each experimental condition. Authors could conclude that astrocytic development is particularly sensitive to the ablation of either one of the two KAT3 proteins, while the mature neuronal phenotype could not be achieved in CBP-ablated NSCs, giving rise instead to differentiation of the oligodendrocyte lineage. By contrast, P300-ablated NSCs remained in pluripotent stages showing mixed lineages, further demonstrating the different fucntions of these two related co-activators/acetyltransferases in the brain. Together, these represent key-concepts in the understanding of KAT3 brain functions and cell differentiation.
I have a main comment on the model that was used as NSC are collected from the SGZ of hippocampal sections from PND 0-3 mouse brains. Even though they are not coming from adult but newborn mice, aren’t they already “specific” progenitor cells compared to those of embryonic neurogenesis ? It seemed to me that the different stages of cell development were different, especially the timed expression of GFAP and may be other (nestin?) markers. Also aren’t these neurons programmed to become granular neurons? I suggest that this model should be better described for example in the introduction, so that the significance of the results can be discussed, i.e. if the described roles for CBP/P300 could be translated to embryonic neurogenesis in the brain (not necessarily in the hippocampus) and thus further in neurological disorders such as Rubinstein-Taybi.
Specific comments
1- Lines 353-355. “Control cultures stained with GFAP exhibited typical stellate with large filamentous morphology while a non-ramified morphology and low level of GFAP immunoreactivity was observed in neurospheres lacking CBP or p300 (Fig. 3A-C).”
Can higher magnifications be shown to illustrate this (i.e. typical stellate versus non-ramified morphology) as this can not be seen with the low focus shown in figure 3A-C. Could GFAP+ cells be quantified? It seems on the image shown that there is less GFAP positive cells in P300-ablated than in CBP or Control neurospheres. Or are there less cells in total? This may be performed with DAPI staining.
2- Figure 4A-C. Labeling with NeuN could be very complementary as it labels the nuclei of mature neurons but without taking into account whether there are some extensions or not, contrary to the TUBB3 labeling.
3- Line 367. CBP “and” P300 should be CBP “or” P300
4- Line 404: CBP instead of CPB.
5- Overall figures are too small and sometimes not of sufficient quality (e.g. Fig 5C) to be read properly.
For example, as we cannot read gene names, each yellow cluster of fig.5C could be clearly identfied by the corresponding cell type names (as cited in the text) written on the side of the heatmap.
6- I do not see any labeling of RGC, OPC or GPC as mentioned in the figure legend for Fig.5D (probably this goes for Figure 6E legend).
7- Lines 401-403. I do not get what is called a “strong deviation” between CBP and P300 ablated populations, regarding cluster 3 (INMN), especially since this is not repeated in the following sentence, which concludes on the main observations of these RNA-seq studies (404-407). May be the percentage of the clusters further investigated (cluster 0/1- cluster 5 and may be also cluster 6) could be given as numbers for each genotype (above the cluster), to clarify this point. For example, looking at the color code, cluster 6 (OL) seems to me to display a strong differential effect between CTRL and CBP, as well as between CBP and P300.
8- Figure 6A-D: should be specified which ones are UP or DOWN on the figure. Figure 6B. Numbers on the fold enrichment axis are not readable. Is it significant?
9- Line 579. “free-floating” neurosphere
Author Response
In this paper, R Gonzalez-Martinez et al. investigated the role of the co-activator/acetyltransferases CBP/P300 (KAT3) in defining the cell lineage from progenitor brain cells. They took advantage of the production of conditional Crebbpf/f, Ep300f/f or Crebbpf/f x Ep300f/f mice to generate neural stem cell (NSC) cultures lacking either one or both proteins. Protein ablation was induced by viral infection driving cre recombinase expression and FACS isolation of specific clones that could be cultured in vitro. Lineage fate was evaluated using different parameters assessing proliferation, differentiation, cell-specific protein expression (immunohistochemistry), and the cell-specific transcriptomics (snRNA-seq). This is a very nice work that, with a “quite simple” and elegant in vitro system, can thoroughly assess and demonstrate important functions for CBP and P300 in cellular fate decisions. Especially the snRNA-seq analyses further provided valuable information about the molecular composition of the produced lineages in each experimental condition. Authors could conclude that astrocytic development is particularly sensitive to the ablation of either one of the two KAT3 proteins, while the mature neuronal phenotype could not be achieved in CBP-ablated NSCs, giving rise instead to differentiation of the oligodendrocyte lineage. By contrast, P300-ablated NSCs remained in pluripotent stages showing mixed lineages, further demonstrating the different functions of these two related co-activators/acetyltransferases in the brain. Together, these represent key-concepts in the understanding of KAT3 brain functions and cell differentiation.
Authors answer (aa): We sincerely thank the reviewer for the kind words and the nice summary of our work. In fact, we have incorporated part of this summary in the revised discussion section (lines 743-749).
I have a main comment on the model that was used as NSC are collected from the SGZ of hippocampal sections from PND 0-3 mouse brains. Even though they are not coming from adult but newborn mice, aren’t they already “specific” progenitor cells compared to those of embryonic neurogenesis? It seemed to me that the different stages of cell development were different, especially the timed expression of GFAP and may be other (nestin?) markers. Also aren’t these neurons programmed to become granular neurons? I suggest that this model should be better described for example in the introduction, so that the significance of the results can be discussed, i. e. if the described roles for CBP/P300 could be translated to embryonic neurogenesis in the brain (not necessarily in the hippocampus) and thus further in neurological disorders such as Rubinstein-Taybi.
aa: The neurospheres used in this study are not from the SGZ of the hippocamus but from the neonatal Subventricular Zone (SVZ) as we stated in the Materials and Methods section (line 97-98). The SVZ is actually a well stablished cornerstone of the neurogenic niche and has been used for many studies as a bona fide source of stem cells. In fact, these cells have been amply characterized in previous studies (Reynolds and Weiss, 1992; Aguirre et al.,2004; Vernerey et al., 2013; Soares et al., 2020) and the percentages of the different cell types obtained from these cells using the differentiation protocol that we follow has also been previously characterized (Reynolds and Weiss, 1992; Ferrón et al., 2009; Belenguer et al., 2016). We provide this information in the introduction section.
Specific comments
1- Lines 353-355. “Control cultures stained with GFAP exhibited typical stellate with large filamentous morphology while a non-ramified morphology and low level of GFAP immunoreactivity was observed in neurospheres lacking CBP or p300 (Fig. 3A-C).” Can higher magnifications be shown to illustrate this (i. e. typical stellate versus non-ramified morphology) as this cannot be seen with the low focus shown in figure 3A-C.
aa: We thank the reviewer for this suggestion. To better visualize the aberrant morphology of CBP and P300 cells, we have now added high magnification pictures of GFAP staining.
Could GFAP+ cells be quantified? It seems on the image shown that there is less GFAP positive cells in P300-ablated than in CBP or Control neurospheres. Or are there less cells in total? This may be performed with DAPI staining.
aa: We have now replaced the pictures in Fig. 3 and the new ones show DAPI staining in blue to better appreciate the changes in GFAP. As the referee noted, it seems that although the number of nuclei is similar in the three types of cultures, there is less GFP signal in the CBP- and P300-ablated cultures than in the controls. As the referee suggested, we have now quantified fluorescence levels of GFAP/nuclei and observed a strong reduction of GFAP staining. We incorporated this quantification to the text (lines 368-369)
2- Figure 4A-C. Labeling with NeuN could be very complementary as it labels the nuclei of mature neurons but without taking into account whether there are some extensions or not, contrary to the TUBB3 labeling.
aa: We appreciate the referee’s suggestion. In fact, we incubated our cultures with NeuN Ab in initial experiments, but soon discovered that the NeuN antobody only labels mature neurons and does not stain the inmature cultured neurons (e.g. Azari et al., PLoS ONE, 2011; Ming and Song, Neuron 2011; Berg et al., F1000Research 2018). This is why we did not detect NeuN positive cells in our cultures by immunocytochemistry methods, nor detected NeuN transcripts (Rbfox3) in the scRNA-seq screen.
3- Line 367. CBP “and” P300 should be CBP “or” P300
aa: This has been corrected
4- Line 404: CBP instead of CPB.
aa: We thank the reviewer for detecting this typo. The mistake has been corrected in several places along the manuscript.
5- Overall figures are too small and sometimes not of sufficient quality (e. g. Fig 5C) to be read properly. For example, as we cannot read gene names, each yellow cluster of Fig. 5C could be clearly identified by the corresponding cell type names (as cited in the text) written on the side of the heatmap.
aa: We thank the reviewer for pointing this out. We have replaced Fig. 5C for a heatmap that includes only the top5 genes of each cluster in order to visualize the names of the genes. The size of the numbers in the X-axis has been increased and the colour-code than in the rest of the figure has been added. We have also included a new table with the top tweenty DEG genes in each cluster (Supplementary material).
6- I do not see any labeling of RGC, OPC or GPC as mentioned in the figure legend for Fig.5D (probably this goes for Figure 6E legend).
Aa: we apologize for this mistake. The referee is correct, these acronyms belong to Figure 6E. This has been corrected.
7- Lines 401-403. I do not get what is called a “strong deviation” between CBP and P300 ablated populations, regarding cluster 3 (INMN), especially since this is not repeated in the following sentence, which concludes on the main observations of these RNA-seq studies (404-407). May be the percentage of the clusters further investigated (cluster 0/1- cluster 5 and may be also cluster 6) could be given as numbers for each genotype (above the cluster), to clarify this point. For example, looking at the color code, cluster 6 (OL) seems to me to display a strong differential effect between CTRL and CBP, as well as between CBP and P300.
aa: We agree that perhaps “strong deviation” was not the best term to describe the alterations observed in cluster 5. “Altered cell distribution” is more acurate. We have replaced it in the manuscript and have also added this observation to the conclusions at the end of the paragraph (lines 406 and 410). In addition, we indicate now the percentage of cells in each cluster (Fig. 5D).
8- Figure 6A-D: should be specified which ones are UP or DOWN on the figure.
Aa: The up and down genes have been now specified in the figure legend.
Figure 6B. Numbers on the fold enrichment axis are not readable. Is it significant?
Aa: We apologize for this. The panel has been replaced and the fold enrichment is now readable. The categories represented in this panel are significant.
9- Line 579. “free-floating” neurosphere
Aa: This has been corrected